# Predictive Water Virology: Hierarchical Bayesian Modeling for Estimating Virus Inactivation Curve

**Syun-suke Kadoya [1], Osamu Nishimura [1], Hiroyuki Kato [2] and Daisuke Sano [1,3,*]**

[1]  Department of Civil and Environmental Engineering, Tohoku University, Aoba 6-6-06, Aramaki, Aoba-ku, Sendai, Miyagi 980-8579, Japan; kadotani9@gmail.com (S.-s.-K.); osamu.nishimura.d2@tohoku.ac.jp (O.N.)

[2]  New Industry Creation Hatchery Center, Tohoku University, Aoba 6-6-10, Aramaki, Aoba-ku, Sendai, Miyagi 980-8579, Japan; h-katou@jiwet.or.jp

[3]  Department of Frontier Science for Advanced Environment, Graduate School of Environmental Studies, Tohoku University, Aoba 6-6-06, Aramaki, Aoba-ku, Sendai, Miyagi 980-8579, Japan

*  Correspondence: daisuke.sano.e1@tohoku.ac.jp; Tel.: +81-22-795-7481

**Abstract:** Hazard analysis and critical control point (HACCP) are a series of actions to be taken to ensure product consumption safety. In food poisoning risk management, researchers in the field of predictive microbiology calculate the values that provide minimum stress (e.g., temperature and contact time in heating) for sufficient microbe inactivation based on mathematical models. HACCP has also been employed for health risk management in sanitation safety planning (SSP), but the application of predictive microbiology to water-related pathogens is difficult because the variety of pathogen types and the complex composition of the wastewater matrix does not allow us to make a simple mathematical model to predict inactivation efficiency. In this study, we performed a systematic review and meta-analysis to construct predictive inactivation curves using free chlorine for enteric viruses based on a hierarchical Bayesian model using parameters such as water quality. Our model considered uncertainty among virus disinfection tests and difference in genotype-dependent sensitivity of a virus to disinfectant. The proposed model makes it possible to identify critical disinfection stress capable of reducing virus concentration that is below the tolerable concentration to ensure human health.

**Keywords:** HACCP; sanitation safety planning; systematic review; enteric viruses; predictive inactivation curve; hierarchical Bayesian model

## 1. Introduction

Wastewater reclamation and reuse have become essential to address the issue of water shortage, but reclaimed and reused wastewater may contain contaminants hazardous to human health. The World Health Organization (WHO) has recommended sanitation safety planning (SSP) for health risk management in wastewater reclamation and reuse, publishing the rules for its use in 2015 [1]. SSP is a scheme for the safe use of excreta, wastewater, and graywater [2]. SSP applied the hazard analysis and critical control point (HACCP) approach to risk management. HACCP was originally developed in the food industry as a tool to prevent food-borne diseases [3], and then the application of HACCP to water management was first suggested in the early 1990s [4,5]. Tsoukalas and Tsitsifli pointed out that the HACCP approach was useful to identify the potential hazards in water treatment processes and was able to determine preventive-corrective actions to reduce the risk to health due to hazardous contaminants [6].

In the HACCP of the food industry, the field of predictive microbiology has played an important role in the management of food-poisoning risk, in which the behavior of pathogens in a food is

quantitatively predicted by mathematical models. Using the results from mathematical models, critical control points (CCPs) are determined to sufficiently inactivate pathogens in a food [3,7]. In heat treatment processes of pathogens in a food or beverage, the proliferation, survival and decay of pathogens are predicted by mathematical models, which determine temperature and duration of heating as CCPs using an inactivation curve estimated in advance. These determined parameters (temperature and time) are monitored in real time and recorded in the HACCP.

The concept of predictive microbiology needs to be applied to predict the behavior of enteric viruses in wastewater (predictive water virology) if SSP is to be applied in wastewater reclamation and reuse. However, the prediction of virus inactivation efficiency has several problems due to the complex composition of the wastewater and the different sensitivity to disinfectants according to virus species. Wastewater quality, such as dissolved organic carbon (DOC), suspended solids (SS), pH, and temperature, can vary among wastewater treatment plants (WWTPs) and even in a WWTP according to the day and season. Difference in wastewater quality can result in varying the efficacy of disinfectant decay and the inactivation efficiency of enteric viruses even in the same disinfectant dose. Furthermore, inactivation efficiency can be different among genotypes/strains of an enteric virus. For example, the chlorine sensitivity of the poliovirus Brunhilde strain was twice that of the Mahoney strain [8,9]. Amarashiri et al. reported genotype-dependent removal of norovirus by membrane filtration because certain genotypes of norovirus combined with the bacteria-bearing histo-blood group antigens [10]. The genotype-dependent inactivation efficiency of enteric viruses needs to be involved in predictive water virology. To manage human health risks, we need to establish the research area of predictive water virology to implement the HACCP approach in microbial risk management in wastewater reclamation and reuse.

In this study, we focused on free chlorine disinfection and tried to construct a predictive inactivation model for virus using information about water quality. We first conducted a systematic review to obtain the inactivation efficiency of enteric viruses (norovirus, rotavirus, hepatitis A virus, adenovirus, coxsackievirus and echovirus) and information about water quality. We used the hierarchical Bayesian model that water qualities were regarded as model variables for training datasets of virus inactivation efficiency, in which a hidden effect of differences among disinfection tests and genotypes of a virus species was taken into account. We then validated the applicability of our model to the datasets newly obtained in this study.

## 2. Materials and Methods

### 2.1. Systematic Review

Beginning in October 2018, we began collecting datasets of inactivation efficiency of enteric viruses by free chlorine according to the framework of systematic review using Google scholar. The keywords used were "virus", "disinfection" and "free chlorine". Dissertations, book chapters, reviews, non-English articles and the articles irrelevant to this study were eliminated from the collected articles. In addition, we eliminated several datasets that showed extremely higher inactivation efficiency in spite of a lower concentration of free chlorine.

### 2.2. Probability Distribution of Inactivation Model Parameters

Datasets of inactivation efficiency expressed as *Log* ($N_t/N_0$) were fitted to the efficiency factor Hom (EFH) model (Equations (1) and (2)) for free chlorine disinfection, which can take the decay of the disinfectant's concentration into account [11,12]. The decay of disinfectants is expressed as the first-order rate equation (Equation (3)).

$$Log\ (N_t/N_0) = -kC_0^n t^m \eta^m, \tag{1}$$

$$\eta = \frac{1 - exp(-nk't/m)}{nk't/m}, \tag{2}$$

$$C_t = C_0 exp(-k't),\tag{3}$$

where $N_t$ is the virus concentration (plaque forming unit, $TCID_{50}$, genome copy number) at contact time $t$ [min], $N_0$ is the initial concentration of virus, $C_0$ is the initial concentration of disinfectant [ppm], $k$ is the inactivation rate constant ($ppm^{-n}\,min^{-m}$), $m$ is the Hom component (-), $n$ is the coefficient of dilution (-), $\eta$ is the efficiency factor (Equation (2)), $C_t$ is the concentration of disinfectant at contact time $t$, and $k'$ is the decay constant ($min^{-1}$). Parameters ($k$, $m$ and $n$) were determined by the least squares method. The Hom component was set as more than 0.3 in the least squares method to ensure an exact effect of the efficiency factor [13]. The decay constants that were unavailable from several reports were estimated using pH, temperature, type of water and initial concentration of disinfectant as parameters by hierarchical Bayesian modeling (HBM) and used as an approximation for estimating EFH model parameters. We first selected several probability distributions for the EFH model based on Akaike's information criterion (AIC) using the "fitdistrplus" package of R software [14]. Hierarchical Bayesian models were constructed using the candidate distributions, and then appropriate probability distributions were selected by comparing the root mean squared error (RMSE) and the widely applicable information criterion (WAIC) [15].

### 2.3. Hierarchical Bayesian Modeling

In this study, population parameters explaining the probability distributions of EFH model parameters were explained by pH, temperature, the type of assay for measuring virus concentration (infectious titer or genome copy number) and the type of water in which the disinfection was conducted (purified water or contaminated water). Categorical variables (type of assay for measuring virus concentration and water) were assigned as 0 (infectivity and purified water) or 1 (genome copy number and contaminated water). The relationship between variables and population parameter was expressed as linear regression using a link function corresponding to the type of probability distribution because if non-linear regression was used, it was difficult to interpret how the squared or cubic variables related to the inactivation efficiency. For example, EFH model parameters followed a log normal distribution, and then an identity link function was utilized after the dataset was converted to logarithmic values (Equation (4)). A logarithmic link function was utilized if an exponential or gamma distribution better fitted Hom or EFH model parameters (Equation (5)).

$$\mu = a_E + b_E \cdot pH + c_E \cdot Temperature + d_E \cdot Assay + e_E \cdot Water,\tag{4}$$

$$Log\ \lambda\ (or\ \beta) = a_E + b_E \cdot pH + c_E \cdot Temperature + d_E \cdot Assay + e_E \cdot Water,\tag{5}$$

where $\mu$ is a logarithmic mean value of an EFH model parameter, $a$–$f$ are coefficients for each variable, $\lambda$ and $\beta$ are rate parameter of gamma and exponential distribution, respectively. We hypothesized that datasets had an experimental error inherent in each study. As shown in Equations (4) and (5), the coefficients were indexed "$E$", indicating that they included an error in each study. The experimental errors were assumed to follow a normal distribution, so that coefficients indexed "$E$" were shown as Equation (6).

$$Coefficient_E \sim Normal\left(\mu_{genotype[i]},\ \sigma_E\right)\tag{6}$$

We also hypothesized the genotype-dependent sensitivity to disinfectants, so the mean value of a coefficient indexed "$E$" was expressed as $\mu_{genotype\,[i]}$. The "$i$" is the index of viral genotypes, and the $\mu_{genotype\,[i]}$ followed a normal distribution (Equation (7)).

$$\mu_{genotype[i]} \sim Normal(\mu_{common},\ \sigma_G)\tag{7}$$

where $\mu_{common}$ is the mean value among all genotypes and $\sigma_G$ is standard deviation bearing the genotype-dependent differences. To evaluate the goodness of fit, the generalized linear model (GLM)

and HBM were evaluated by RMSE and WAIC. Both model constructions were executed on statistical software R (version 3.5.0, R Foundation for Statistical Computing, Vienna, Autstria) by R and Stan codes. (Tables S1, Code S1 and S2).

*2.4. Model Validation*

2.4.1. Measurement of Norovirus Concentration in Treated Wastewater

Constructed models were validated using data of inactivation efficiency obtained by norovirus concentration in treated wastewater of a WWTP. Wastewater samples (four sampling points in a disinfectant pool) were collected on 4 February 2019 in Sendai, Japan. Noroviruses in wastewater sample were concentrated by electronegative membrane as previously described by Haramoto et al. [16]. Murine norovirus (strain S7-PP3) stock as a process control was added to wastewater samples to evaluate recovery efficiency. Genome copy number of norovirus in concentrate was measured by RT-qPCR using a Taqman probe (nucleotide sequences of forward/reverse primers and probe outlined by Kageyama et al.) [17]. The initial free chlorine concentration was 2.0 ppm, and the decay constant was estimated to be 17.97 (min$^{-1}$).

2.4.2. Inactivation Experiment of Rotavirus by Sodium Hypochlorite

Rhesus rotavirus (RRV; G3P[3]) was used for free chlorine inactivation in our laboratory. The free chlorine solution was prepared by diluting sodium hypochlorite (Wako Pure Chemical Industries, Ltd., Osaka, Japan) with Dulbecco's PBS (-) (Nissui Pharmaceutical Co., Ltd., Tokyo, Japan). The RRV (1.0 mL) suspension was added to 9.0 mL of the free chlorine solution. At each time step (0, 5–15 s; 1, 2, 5 and 10 min), 1.0 mL of each mixture was picked up and neutralized by 1.0% sodium thiosulfate. As described in our previous report, an infectious titer of RRV was measured by plaque assay [18]. Free chlorine concentration was measured at each time step, and the decay constant $k'$ was estimated to be 7.01 (min$^{-1}$). In this experiment, the initial concentration of free chlorine was set at 2.27 ppm.

2.4.3. Predictive Inactivation Curve

The predictive inactivation curve of the norovirus was determined using variables relating to water quality in a WWTP in Sendai (pH = 6.65, temperature = 16.6, contaminated water and genome copy number). In the case of rotavirus, water-quality variables were set as follows: pH = 7.0, temperature = 20, contaminated water and infectivity. For other viruses, several datasets (test datasets) were used for model validation. The number of test datasets was more than 10% of the total datasets for each virus species. Predictive inactivation curves were based on the water quality of test datasets by using the constructed models and then compared to test datasets. RMSEs were calculated to evaluate the goodness of fit for test datasets. R code for a predictive inactivation curve was described in a supplementary material (Code 3).

## 3. Result

*3.1. Article Selection and Data Extraction*

We first identified 4174 records on the Web using the keywords "virus", "disinfection" and "free chlorine" (Figure 1). We then excluded dissertations, book chapters, reviews, conference reports and non-English papers (first screening), and the number of articles decreased to 2184. By excluding articles irrelevant to our study and inappropriate datasets (e.g., only two time points for measuring an inactivation efficiency, a low level of free chlorine concentration but higher inactivation efficiency in a short time compared to other datasets) (second screening), 29 full-text articles relevant to free chlorine disinfection for viruses remained.

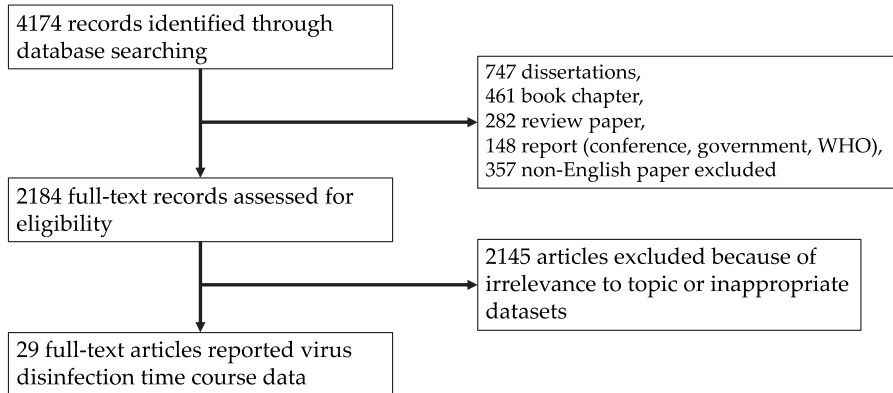

**Figure 1.** The flow of article screening. Twenty-nine full-text articles relating to free chlorine were selected.

The number of training datasets for norovirus, rotavirus, poliovirus, hepatitis A virus, adenovirus, coxsackievirus, and echovirus was 12, 25, 35, 9, 16, 29, and 12, respectively [19–46]. The test datasets were selected to make the number of datasets of each article as uniform as possible. The number of test datasets of poliovirus, hepatitis A virus, adenovirus, coxsackievirus and echovirus was 31, 1, 8, 6, and 3, respectively.

*3.2. Probability Distribution of the Inactivation Model Parameters*

The probability distributions of EFH model parameters optimal for predictive inactivation models were determined by comparing WAIC and RMSE for HBM among lognormal, gamma, Weibull and exponential distributions. Almost all parameters of the EFH model followed lognormal distributions, but both inactivation rate constant *k* and coefficient of dilution *n* of echovirus followed exponential distributions (Table 1). Only the Hom component *m* of poliovirus followed a gamma distribution.

**Table 1.** Probability distribution for parameters of the efficiency factor Hom (EFH) model.

|  | k | m | n |
| --- | --- | --- | --- |
| Norovirus | Lognormal | → * | → |
| Rotavirus | Lognormal | → | → |
| Poliovirus | Lognormal | Gamma | Lognormal |
| Adenovirus | Lognormal | → | → |
| Hepatitis A virus | Lognormal | → | → |
| Coxsackievirus | Lognormal | → | → |
| Echovirus | Exponential | Lognormal | Lognormal |

* the probability distribution was same as left one.

*3.3. Comparison of a Goodness of Fit between GLM and HBM*

In the models of *k* (Table 2), both WAIC and RMSE values of HBM were lower than those in GLM. More than half of virus species showed lower WAIC for the *m*-value of GLM, but RMSEs of HBM were equal to or smaller than those of GLM. RMSEs of HBM for *n* were also smaller than GLM, but some values of WAIC of GLM (such as hepatitis A virus) were smaller than those of HBM. The predicted values of several parameters were better fitted to the observed values than to GLM, and the 95% confidence intervals of predicted EFH parameters of HBM were narrower (*k* and *n* of norovirus, *m* of hepatitis A virus and all parameters of rotavirus, poliovirus and adenovirus) (Figure 2). In the same manner as WAIC and RMSE, the predicted EFH parameters for coxsackievirus and echovirus seemed to be the same between GLM and HBM.

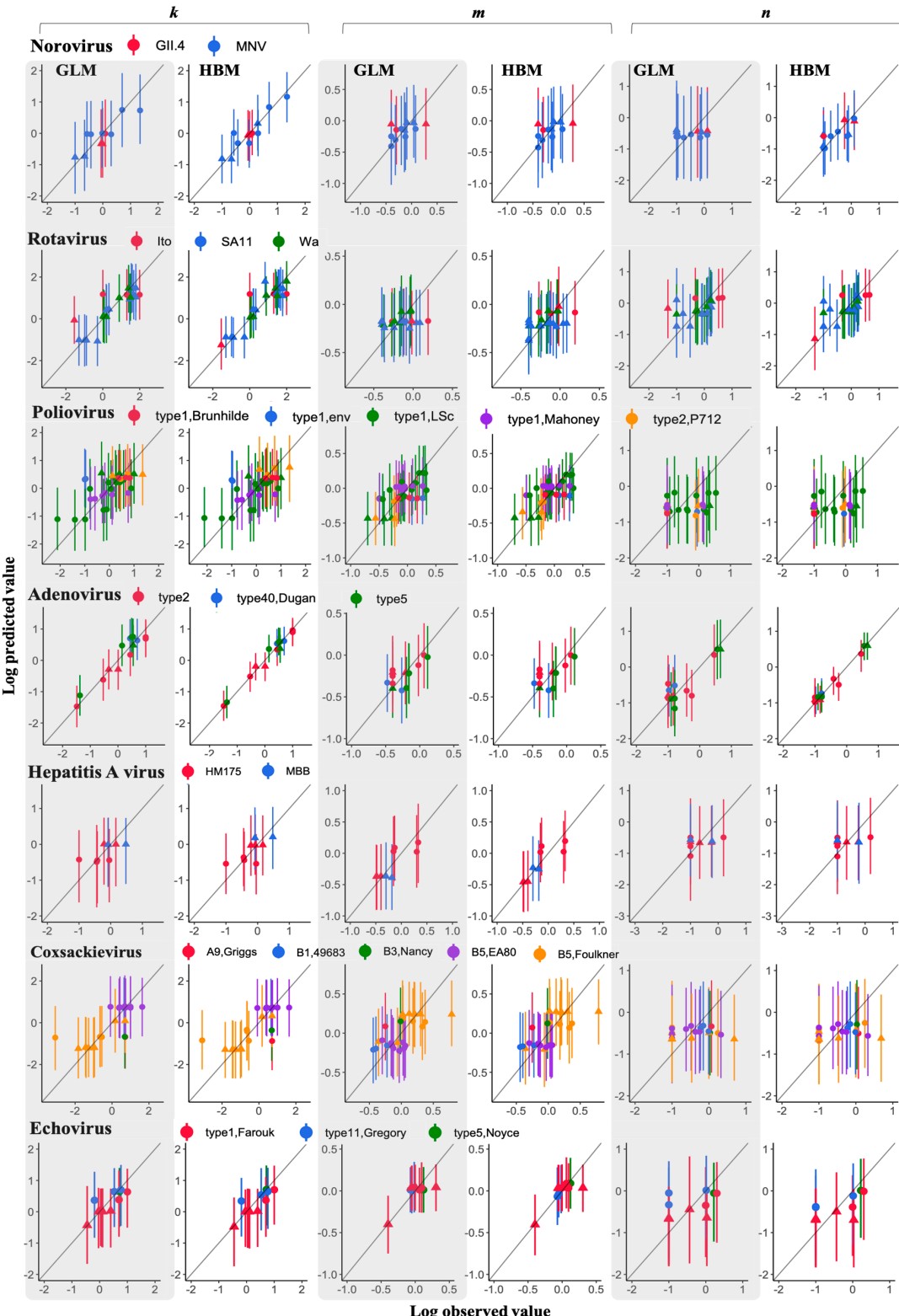

**Figure 2.** Comparison of a goodness of fit between the generalized linear model (GLM) and hierarchical Bayesian model (HBM) for EFH model parameters (circle: purified water, triangle: contaminated water). Each color shows the type of viral genotypes, and the gray shade indicates the results of GLM.

**Table 2.** Comparison of a goodness of fit between generalized linear model (GLM) and hierarchical Bayesian model (HBM).

|  | | k | | m | | n | |
|---|---|---|---|---|---|---|---|
|  | Statistics | GLM | HBM | GLM | HBM | GLM | HBM |
| Norovirus | WAIC | 23.0 | 14.4 | 6.81 | 7.54 | 28.9 | 17.4 |
|  | RMSE | 0.34 | 0.23 | 0.03 | 0.03 | 0.43 | 0.27 |
| Rotavirus | WAIC | 54.6 | 45.8 | −7.90 | −11.2 | 43.9 | 35.6 |
|  | RMSE | 0.45 | 0.41 | 0.16 | 0.14 | 0.20 | 0.14 |
| Poliovirus | WAIC | 69.5 | 68.8 | 37.9 | 36.3 | 63.9 | 63.3 |
|  | RMSE | 0.54 | 0.53 | 0.51 | 0.49 | 0.50 | 0.49 |
| Adenovirus | WAIC | 11.0 | 1.77 | −3.12 | −5.92 | 17.9 | −2.19 |
|  | RMSE | 0.05 | 0.01 | 0.02 | 0.02 | 0.08 | 0.01 |
| Hepatitis A virus | WAIC | 19.4 | 11.9 | 4.02 | 2.04 | 18.1 | 18.4 |
|  | RMSE | 0.29 | 0.27 | 0.17 | 0.15 | 0.16 | 0.15 |
| Coxsackievirus | WAIC | 67.9 | 65.5 | 2.21 | 3.98 | 48.1 | 47.5 |
|  | RMSE | 0.70 | 0.68 | 0.20 | 0.19 | 0.51 | 0.49 |
| Echovirus | WAIC | 50.6 | 48.1 | −9.30 | −8.13 | 18.1 | 13.5 |
|  | RMSE | 0.24 | 0.23 | 0.10 | 0.09 | 0.46 | 0.41 |

### 3.4. Predictive Inactivation Curves Based on HBM

The norovirus inactivation curve for free chlorine was predicted using the dataset of norovirus (genogroup II: GII) inactivation and water quality obtained from a WWTP in Japan on February 2019 (Figure 3a). Predictive inactivation curves were depicted using the 2.5, 25, 50, 75 and 97.5% values of $k$, $m$ and $n$. Except for the 97.5% curve, predictive inactivation curves converged around −0.3. Real data (circle plots in Figure 3a) of norovirus inactivation efficiency were on the curve using the 25 or 50% values. The RMSE of the 50% curve and test data was 0.02. The inactivation efficiency of the rhesus rotavirus (RRV) strain was obtained by an inactivation experiment in our laboratory. The predictive curve of rotavirus inactivation in a stationary phase ranged from −0.2 to −1.8 (Figure 3b). Experimental data were within the 95% confidence interval of the predictive curve and the RMSE of the 50% curve and test data was 0.38.

In the case of other virus species, test datasets from literatures were compared to the predictive inactivation curves generated by HBM (Figure 4). The $m$-value in the EFH model generally takes from 0.3 to 2.5 [12], which provides the appropriate value of the efficiency factor. The predictive curves of which the $m$-value did not take the appropriate range are shown as dashed lines in Figure 4. In the first dataset of poliovirus, until 3 min, test datasets did not deviate from the predictive curve (Figure 4a; 0.29 ppm of free chlorine, $k' = 0.001$ (min$^{-1}$), pH = 7.0, temperature = 26.5 °C, infectivity and purified water [47]), whereas the second dataset were in the range of prediction after 1.5 min of contact time (Figure 4b; 0.49 ppm of free chlorine, $k' = 0.001$ (min$^{-1}$), pH = 7.8, temperature = 5.0 °C, infectivity and purified water [28]). The test data of adenoviruses gathered near the 50% predictive curve (Figure 4c; 5.4 ppm of free chlorine, $k' = 3.13$ (min$^{-1}$), pH = 8.0, temperature = 25 °C, infectivity and purified water [24]). The predictive curve of hepatitis A virus converged to about −1.3, and test data were on the predictive curve except for a value of inactivation efficiency at 2 min (Figure 4d; 0.5 ppm of free chlorine, $k' = 1.17$ (min$^{-1}$), pH = 6.0, temperature = 5.0 °C, infectivity and purified water [37]). The predictive curves of coxsackievirus displayed a weaker tailing, so that the range of the predictive curve widened as time went on (Figure 4e; 0.25 ppm of free chlorine, $k' = 0.07$ (min$^{-1}$), pH = 10, temperature = 26.5 °C, infectivity and purified water [47]). The test data of coxsackievirus were placed from 25% to 50% predictive curves. The test data of echovirus were on the center of the predictive inactivation curve (Figure 4f; 0.2 ppm of free chlorine, $k' = 0.47$ (min$^{-1}$), pH = 7.0, temperature = 5.0 °C, infectivity and contaminated water [34]). The RMSEs of from (a) to (f) were 0.95, 0.58, 0.52, 0.61, 0.84 and 1.02, respectively.

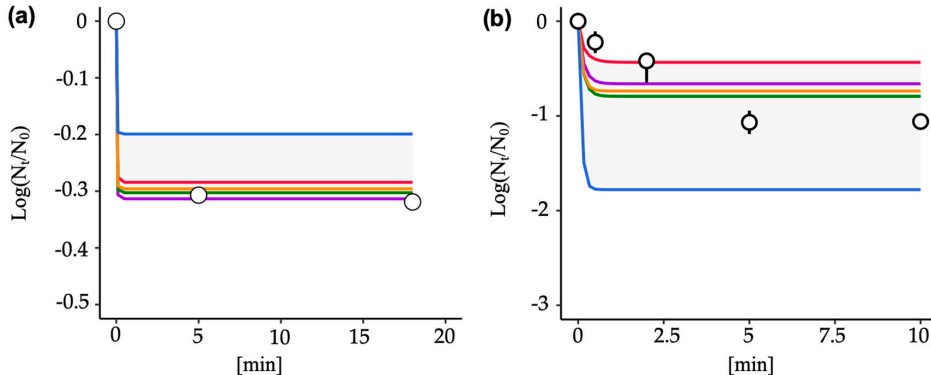

**Figure 3.** Predictive inactivation curves of norovirus GII and rhesus rotavirus by free chlorine using real datasets (red: 2.5%, purple: 25%, green: 50%, orange: 75%, blue: 97.5% curves). Circle plots are test data of inactivation efficiency. (**a**) A predictive curve for norovirus GII was generated under the following conditions: 2.0 ppm of free chlorine, $k' = 17.9$ (min$^{-1}$), pH = 6.74, temperature = 16.6 °C, genome copy number and contaminated water. (**b**) A predictive curve for a rhesus rotavirus strain was generated under the following conditions: 2.27 ppm of free chlorine, $k' = 7.01$ (min$^{-1}$), pH = 7.0, temperature = 20 °C, infectivity and contaminated water.

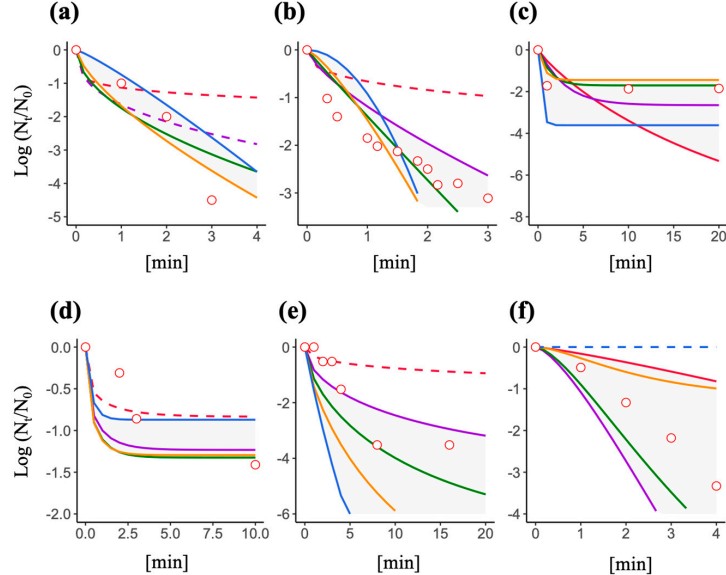

**Figure 4.** Validation of predictive inactivation models for free chlorine using test datasets (red: 2.5%, purple: 25%, green: 50%, yellow: 75%, blue: 97.5% curve). (**a**) Poliovirus (type 1 Mahoney) inactivation under the condition; 0.29 ppm of free chlorine, $k' = 0.001$ (min$^{-1}$), pH = 7.0, temperature = 26.5 °C, infectivity and purified water [47], (**b**) Poliovirus (type 1 Mahoney) inactivation under the condition; 0.49 ppm of free chlorine, $k' = 0.001$ (min$^{-1}$), pH = 7.8, temperature = 5.0 °C, infectivity and purified water [28], (**c**) Adenovirus (type 2) inactivation under the condition; 5.4 ppm of free chlorine, $k' = 3.13$ (min$^{-1}$), pH = 8.0, temperature = 25 °C, infectivity and purified water [24], (**d**) Hepatitis A virus (HM175) inactivation under the condition; 0.5 ppm of free chlorine, $k' = 1.17$ (min$^{-1}$), pH = 6.0, temperature = 5.0 °C, infectivity and purified water [37], (**e**) Coxsackievirus (B5) inactivation under the condition; 0.25 ppm of free chlorine, $k' = 0.07$ (min$^{-1}$), pH = 10, temperature = 26.5 °C, infectivity and purified water [47], (**f**) Echovirus (type 1 Farouk) inactivation under the condition; 0.2 ppm of free chlorine, $k' = 0.47$ (min$^{-1}$), pH = 7.0, temperature = 5.0 °C, infectivity and contaminated water [34]. The dashed lines indicate predictive curves that are inappropriate to use because the *m*-values were less than 0.3.

### *3.5. Effect of a Decay Constant and Efficiency Factor on the Shape of a Predictive Inactivation Curve*

A decay constant $k'$ can influence the shape of a predictive inactivation curve. In the case that the decay constant was 10, a gradual tailing in curves was seen (Figure 5a). As the value of the decay constant increased to more than 17.9, the predictive curves became steep, and the inactivation efficiency decreased (Figure 5b,c) in spite of the same condition for disinfection except for a decay constant.

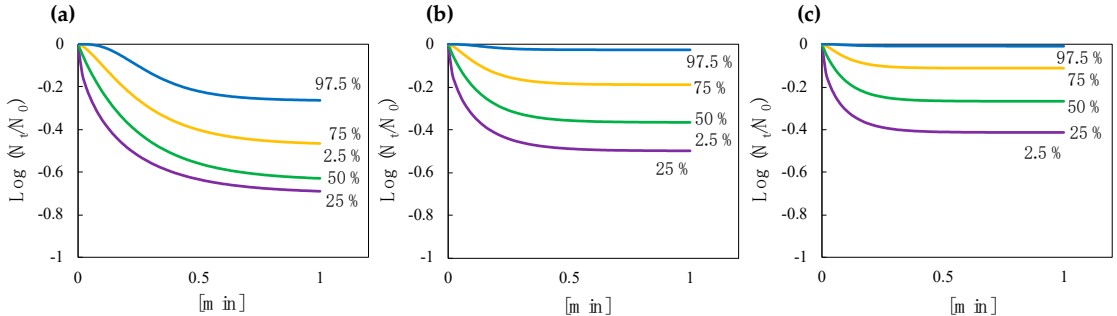

**Figure 5.** Predicted inactivation curves constructed by different $k'$-values (norovirus inactivation at the condition: 2.0 ppm of free chlorine, Ph = 7, temperature = 20 °C, genome copy number and contaminated water). (**a**) $k' = 10$, (**b**) $k' = 17.9$, (**c**) $k' = 25$.

## 4. Discussion

We have constructed predictive inactivation models for free chlorine disinfection of enteric viruses by hierarchical Bayesian modeling to estimate the EFH model parameters ($k$, $m$, $n$). Datasets about inactivation efficiency and water qualities were extracted from 29 full-text articles selected by the framework of systematic review (Figure 1). Almost all EFH model parameters followed a lognormal distribution except for the $k$-value of echovirus, the $m$-value of poliovirus and the $n$-value of poliovirus and echovirus (Table 1). HBMs of each virus species considering the effect of an uncertainty among disinfection tests and genotype-dependent sensitivity (unexplained in literatures) to disinfectants mostly showed better goodness of fit than GLM (Table 2 and Figure 2). Predictive inactivation models based on HBM could explain experimental (norovirus and rotavirus) and test datasets (other viruses), but several test datasets deviated from the prediction curves (Figures 3 and 4). We then confirmed an effect of a decay constant $k'$ of free chlorine on a predictive inactivation curve under the same conditions of disinfection (Figure 5).

Hierarchical Bayesian modeling (HBM) mostly showed better predictions of the EFH model parameters than generalized modeling (GLM) (Table 2 and Figure 2), which implies the existence of genotype-dependent sensitivity unexplained in each investigation and/or that of uncertainty (e.g., experimental error or skill) among disinfection tests. Genotype-dependent sensitivity of viruses has been reported [8,9,47,48], and also a changeable sensitivity to disinfectants within a genotype is currently being investigated [49–51]. Enteric viruses (especially, RNA viruses) exist as a genetically diverse population (termed "quasispecies") due to the higher mutation rate [52,53]. Among laboratories, in addition to the skill of researchers or the precision of experiments, the population structures of test viruses can be different among studies, which can result in different sensitivities to disinfectants among studies. The different population structure of a virus strain can bear the difference in inactivation efficiency among disinfection tests despite same virus strain or similar conditions for disinfection. On the other hand, several EFH model parameters estimated by HBM displayed less goodness of fit than those estimated by GLM. For example, the WAIC of m-values predicted by GLM for norovirus, coxsackievirus and echovirus were smaller than those by HBM (Table 2), which could be attributed to the variability of values of water-quality variables being too small (such as pH and temperature) for uncertainty among disinfection tests and genotype-dependent sensitivity to explain the inactivation model parameters.

Our predictive inactivation models predicted real datasets better, but did not explain several test data points (Figures 3 and 4). In this study, continuous variables were only pH and temperature while categorical variables were water types (purified or contaminated water) and the types of viral titer measurement (infectivity or genome copy number). Water quality in a practical treated wastewater should be explained by DOC, SS and ammonia nitrogen (AN), but water-quality datasets were limited in literatures despite regarding the selection of variables as important [54,55]. To solve this problem about the small number of variables, we paid particular attention to the decay constant $k'$.

The decay constant $k'$ can impact the form of predictive inactivation curves as shown in Figure 5. Since the concentration of chemical disinfectants continues to decrease due to the consumption by contaminants such as DOC, SS and AN [56], the disinfectant decay can be explained by the levels of such contaminants in wastewater. The level of water quality changes corresponding to the transition of seasons, which results in the seasonal variation of the decay constant. We suggest that instead of applying extra variables of water quality to inactivation parameters, the disinfectant decay $k'$ should be surveyed and monitored at each WWTP in order to precisely predict the inactivation efficiency.

We then considered which predicted curves should be used as a criterion that can sufficiently prevent the risk of enteric virus infection. Test datasets tended to get close to the 50% predictive curves (Figures 3 and 4). Since the 50% predictive curve can overestimate the inactivation efficiency and underestimate human health risks, the curve above 50% predictive curve should be used as a criterion of inactivation efficiency in each WWTP.

Our study indicated that predictive water virology using HBM is useful to determine critical limits (disinfectant concentration and contact time) for the critical control point at each WWTP. Prediction of virus concentration in treated wastewater by our model can be useful to estimate the health risk to determine the critical limits. The United States Environmental Protection Agency (USEPA) and WHO suggested tolerable risk and disease burden as $10^{-4}$ infection/person/year and $10^{-6}$ ($10^{-4}$ and $10^{-5}$ as less stringent setting) DALY loss per person per year (DALYpppy loss) [57,58]. In the case of applying DALY, according to values of tolerable DALY adopted at each WWTP, free chlorine concentration and its contact time as critical limits must be determined (Figure 6). Also, risks estimated by predictive inactivation models are different among virus species, so that operators at each WWTP should determine critical limits to enable the estimated risks of all viruses to go below the employed tolerable risk or disease burden (Figure 6).

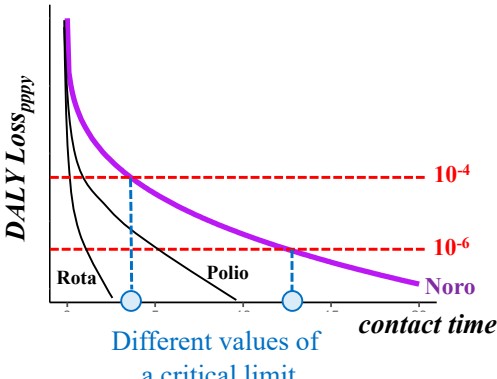

**Figure 6.** Example curves of DALYpppy loss for three virus species. Blue circles indicate the critical limits at each value of acceptable disease burden suggested by WHO [2]. Critical limits are the intersection point of the x-axis and DALYpppy loss curve for norovirus, which shows a higher burden of disease than other viruses.

## 5. Conclusions

We established the predictive water virology to predict the virus inactivation efficiency in wastewater in order to fit into the SSP (HACCP) approach. To construct the predictive inactivation

model, hierarchical Bayesian modeling was used for the prediction of EFH model parameters with water-quality data and mostly showed better prediction than generalized linear modeling, which implied the existence of genotype-dependent sensitivity to free chlorine and uncertainty among virus disinfection tests. We found that test datasets tended to gather the 50% predictive curves and the decay constant played an important role in forming predictive inactivation curves. Therefore, we suggest that disinfectant decay *k′* should be measured at each WWTP to compensate for the small number of variables, and the curve above the 50% predictive curve should be used as a criterion to ensure human health. Our predictive inactivation models enable WWTPs to contribute to the improvement of public health related to water-related viruses by determining the appropriate setting of disinfection stress that can be monitored in the HACCP framework.

**Supplementary Materials:** The following are available online at http://www.mdpi.com/2073-4441/11/10/2187/s1, Table S1: Norovirus data analyzed for model construction, Code S1: R code for model construction, Code S2: Stan code for model construction, Code S3: R code for the prediction of EFH model parameters.

**Author Contributions:** Conceptualization, S.-s.K. and D.S.; Methodology, S.-s.K.; Validation, O.N., H.K. and D.S.; Formal Analysis, S.-s.K.; Investigation, S.-s.K.; Data Curation, S.-s.K.; Writing-Original Draft Preparation, S.-s.K.; Writing-Review & Editing, H.K., D.S. and O.N.; Visualization, S.-s.K.; Supervision, D.S.; Project Administration, D.S.

**Funding:** This study was supported by Gesuido Academic Incubation to Advanced Project, Japan Ministry of Land, Infrastructure, Transport and Tourism, and "The Sanitation Value Chain: Designing Sanitation Systems as Eco-Community Value System" Project, Research Institute for Humanity and Nature (RIHN, Project No.14,200,107).

**Acknowledgments:** The authors thank the Sen-En Joka center, a wastewater treatment plant in Sendai, Japan to provide wastewater samples.

**Conflicts of Interest:** We do not have any conflicts of interest in this study.

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
