# Peer review of "Predictive Water Virology: Hierarchical Bayesian Modeling for Estimating Virus Inactivation Curve"

_water, doi:10.3390/w11102187_

Round 1
Reviewer 1 Report
The manuscript is extensive and interesting for readers. However, language and style needs improvement. Further review is not necessary.
Author Response
(comment1)
The manuscript is extensive and interesting for readers. However, language and style needs improvement. Further review is not necessary.
(Reply1)
Thank you for a kind comment. We revised English sentences of a manuscript.
Reviewer 2 Report
1) Why do the authors use linear regression methodologies? Isn't there a specific theory to provide more physically-related
2) Besides the graphs can the authors please provide summary fit measures?
Author Response
(comment1)
Why do the authors use linear regression methodologies? Isn't there a specific theory to provide more physically-related.
(Reply1)
Non-linear regression can improve fit parameters but overfitting to training data is found. Also, the intuitive understanding of model by linear regression is better than non-linear regression because it is difficult to interpret how squared (or cubic) variables relates to virus inactivation efficiency. (Line 120-123)
(comment2)
Besides the graphs, can the authors please summary fit measures?
(Reply2)
Thank you for a good suggestion. We calculated RMSE of test datasets for 50% predictive inactivation curves. (Line 177-178, 236, 239-240, 269-270)
Reviewer 3 Report
The study employed the hierarchical Bayesian modeling to estimate efficiency factor Hom (EFH) model parameters. The model is validated with experimental and literature data. The study indicates the HBM has better prediction than generalized linear model for parameter estimation. Overall the paper is clear presented. I found the paper valuable to read. I would suggest the publication of the paper.
Author Response
(comment1)
The study employed the hierarchical Bayesian modeling to estimate efficiency factor Hom (EFH) model parameters. The model is validated with experimental and literature data. The study indicates the HBM has better prediction than generalized linear model for parameter estimation. Overall the paper is clear presented. I found the paper valuable to read. I would suggest the publication of the paper.
(reply1)
Thank you for your kind comment. We corrected English sentences of a manuscript.
Round 2
Reviewer 2 Report
Accept as is, very good work